# Distinct risk groups with different healthcare barriers and acute care use exist in the U.S. population with chronic liver disease

Carrie R. Wong[1,2,3,4]*, Catherine M. Crespi[2,3,5], Beth Glenn[2,3,4], Steven-Huy B. Han[1], James A. Macinko[2,4,6], Roshan Bastani[2,3,4]

**1** Vatche and Tamar Manoukian Division of Digestive Diseases, Department of Medicine, University of California, Los Angeles, California, United States of America, **2** Kaiser Permanente Center for Health Equity, University of California, Los Angeles, California, United States of America, **3** Jonsson Comprehensive Cancer Center, University of California, Los Angeles, California, United States of America, **4** Department of Health Policy and Management, Fielding School of Public Health, University of California, Los Angeles, California, United States of America, **5** Department of Biostatistics, Fielding School of Public Health, University of California, Los Angeles, California, United States of America, **6** Department of Community Health Sciences, Fielding School of Public Health, University of California, Los Angeles, California, United States of America

* crwong@mednet.ucla.edu

## Abstract

### Background

The relationship between community-based healthcare barriers and risk of recurrent hospital-based care among persons with chronic liver disease (CLD) is understudied. We aimed to uncover distinct groups among adults in the United States with CLD based on healthcare barriers and risk-stratify recurrent acute care use by group.

### Methods

Using National Health Interview Survey (2011 to 2017) data, we performed latent class analysis (LCA) to uncover groups experiencing distinct sets of healthcare barriers. We assessed sociodemographic and health characteristics and probabilities of recurrent acute care use by group.

### Results

The sample included 5,062 (estimated 4.7 million) adults with CLD (median [range] age 55 [18–85]). LCA modeling differentiated 4 groups: *minimal barriers* (group 1) (n = 3,953; 78.1%), *unaffordability* (group 2) (n = 540; 10.7%), *care delays* (group 3) (n = 328; 6.5%), and *inability to establish care* (group 4) (n = 240; 4.8%). Group 2 had the most uninsured persons (n = 210; 38.9%), whereas group 3 was mostly insured (n = 305; 93.1%). Group 4 included the most adults under 65 years old (n = 220; 91.7%), females (n = 156; 65.1%), and persons with unemployment (n = 169; 70.6%) and poverty (n = 85; 35.3%). Compared to group 1, the likelihood of recurrent acute care use was highest for group 4 (aOR, 1.85;

**Funding:** This study was supported by the Ruth L. Kirschstein-National Research Service Award (HRSA T32-HP19001) training grant and the AASLD Foundation Advanced/Transplant Hepatology Fellow Award to Carrie Wong. Funders did not play any role in the study design, data collection, analysis, decision to publish, or preparation of the manuscript.

**Competing interests:** The authors have declared that no competing interests exist.

95% CI, 1.23–2.79 followed by group 3 (aOR, 1.50; 95% CI, 1.07–2.11) and group 2 (aOR, 1.48; 95% CI, 1.11–1.97).

## Conclusion

US adults with CLD can be categorized into 4 distinct groups based on healthcare barriers, which are associated with different probabilities of recurrent acute care use. Findings from this study are important for future interventions to reduce potentially avoidable hospital-based care among the highest-risk persons with CLD.

## Introduction

Rates of hospitalizations have disproportionately increased for individuals with chronic liver disease (CLD) [1–3]. Accordingly, there have been efforts to predict and reduce hospitalizations for persons with CLD; however, prior work has shown modest predictive accuracy (C-statistics 0.60–0.75) [4–9]. These predictive algorithms have largely focused on hospital-based clinical variables which may have limited their predictive yield after hospital discharge.

Attention to non-hospital-based factors, specifically one's experience with healthcare barriers in the community setting, can provide new knowledge on the potential influence of outpatient access to care on recurrent acute care use. Care coordination programs have shown that frequent interactions with providers in the ambulatory setting [10] and post-discharge home visits [11–13] have successfully reduced rehospitalizations, potentially through the elimination of some healthcare barriers in the care seeking process.

Emerging data have shown that specific healthcare barriers, including healthcare unaffordability [14] and transportation insecurity [15], among adults in the United States (US) with CLD are associated with increased likelihood of acute care use. However, the relative risk of different healthcare barriers on recurrent acute care use and the sociodemographic and health characteristics of persons who share unique patterns of barriers among the US adult population with CLD remain unknown.

To our knowledge, identification of distinct risk groups for recurrent acute care in the US adult population with CLD using community-based healthcare barriers has not been done. In this study, we used National Health Interview Survey (NHIS) data to perform latent class analysis (LCA) to uncover risk groups based on self-reported healthcare barriers among respondents with CLD. We then evaluated the likelihood of latent class (risk group) membership by sociodemographic, health, and insurance characteristics and measured the probability of recurrent acute care use by risk group.

## Methods

### Data source

We pooled annual NHIS data, from 2011 to 2017, to conduct a cross-sectional study that yielded nationally representative estimates from noninstitutionalized, community-dwelling persons residing in the 50 states and District of Columbia at the time of the interview [16]. NHIS is an annual in-person household interview that collects data about sociodemographic and health status [16]. Study years were selected to capture healthcare experiences after the enactment of the Affordable Care Act. NHIS no longer surveyed specific healthcare barriers in 2018. We used the Sample Adult file, which included responses from randomly selected adults

per randomly selected household. During this study period, the mean conditional response and final response rates were 80.7% and 60.4%, respectively.

The institutional review board at the University of California, Los Angeles exempted this study from review for the use of de-identified and publicly available data from the Integrated Public Use Microdata Series Health Surveys [17]. The study follows the Strengthening the Reporting of Observational Studies in Epidemiology guidelines.

### Study population

Our study population consisted of respondents aged 18 years or older with CLD, which included persons who responded *yes* to the questions, "Has a doctor or other health professional ever told you that you had any kind of chronic, or long-term liver condition" or "During the past 12 months, have you been told by a doctor or other health professional that you had any kind of liver condition?" as previously done [14, 15] (Fig 1).

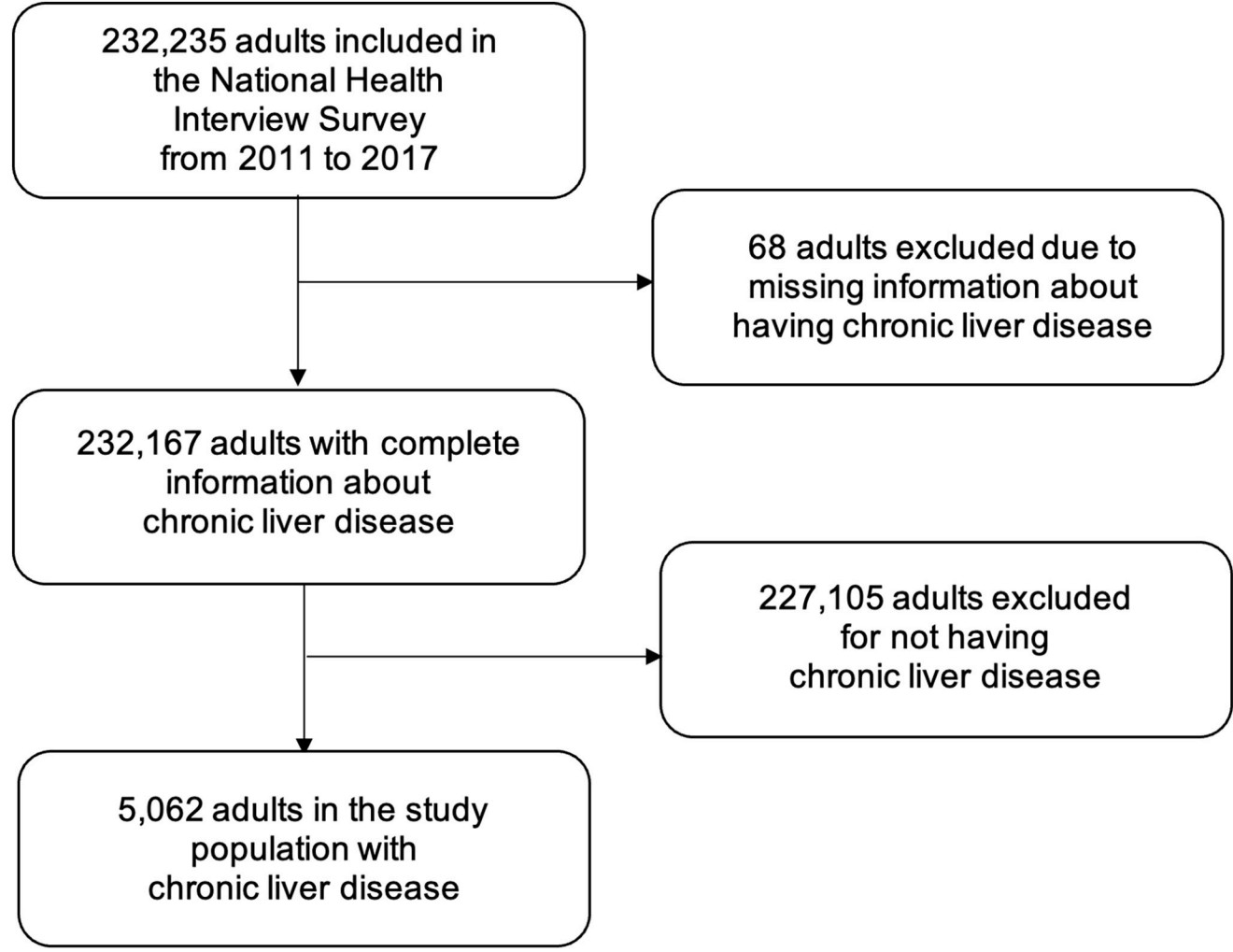

**Fig 1. Study population flowchart.** Chronic liver disease was based on *yes* responses to the following questions: "Has a doctor or other health professional ever told you that you had any kind of chronic, or long-term liver condition?" or "During the past 12 months, have you been told by a doctor or other health professional that you had any kind of liver condition?".

## Measures

We selected self-reported healthcare barriers that spanned the care seeking process including organizational barriers and healthcare unaffordability. Respondents were considered to have a healthcare barrier if they responded *yes* to any of the following questions in the past year:

1. "Were you told by a doctor's office or clinic that they would not accept you as a new patient?"

2. "Were you told by a doctor's office or clinic that they did not accept your healthcare coverage?"

3. "Did you have any trouble finding a general doctor or provider who would see you?"

4. "Was there any time when you needed medical care but did not get it because of the cost?"

5. "Was there any time when you needed any of the following, but didn't get it because you couldn't afford it . . .follow-up care?"

6. "Was there any time when you needed any of the following, but didn't get it because you couldn't afford it . . .to see a specialist?"

7. "Was there any time when you needed any of the following, but didn't get it because you couldn't afford it . . .prescription medicines?"

8. "Have you delayed getting care for any of the following reasons in the past 12 months? . . .You didn't have transportation?"

9. "Have you delayed getting care for any of the following reasons in the past 12 months? . . .You couldn't get an appointment soon enough?"

10. "Have you delayed getting care for any of the following reasons in the past 12 months? . . .The clinic/ doctor's office wasn't open when you could get there?"

11. "Have you delayed getting care for any of the following reasons in the past 12 months? . . .You couldn't get through on the telephone?"

12. "Have you delayed getting care for any of the following reasons in the past 12 months? . . .Once you get there, you have to wait too long to see the doctor?"

We also included any respondents who responded *no* to the following question:

13. "Is there a place that you usually go to when you are sick or need advice about your health?"

We assessed a parsimonious set of covariates about the respondents' sociodemographic, health, and insurance status based on reported associations between the covariates and health-care experiences: age, sex, race or ethnicity (Hispanic, Non-Hispanic [NH] White, NH Black/ African-American, NH Asian, NH American Indian/ Alaskan Native, NH Other) [18, 19], number of comorbidities [20, 21], functional limitation due to health [22, 23], fair or poor health (vs. excellent, very good, or good health) [24, 25], education attainment (less than vs. at least high school graduate level education) [26], employment (working vs. not working in the past year) [27], household poverty (less than or at least 200% of the federal poverty level [FPL]) [16], household support (i.e. living alone) [28], and health insurance, which included public (Medicaid, other state-sponsored insurance), Medicare (without Medicaid or other state-spon-sored insurance), private, or none [29, 30].

We included respondents' report of recurrent acute care use defined as having at least two emergency department (ED) visits and/or overnight hospital admissions for any cause in the

past year. We did not limit hospital-based acute care to readmissions because up to 40% of recurrent acute care encounters within 30 days of a hospital discharge could occur in the ED without readmission [31].

## Statistical analysis

We conducted LCA to uncover unique groups of persons with CLD in the US population. These groups are the latent classes, which are derived from patterns and probabilities of responses to a set of variables (i.e. questions about different healthcare barriers) [32] (Fig 2).

We used a sequential exploratory approach by fitting different models with a successively increasing number of latent classes starting with one class ($k$ = 1, 2, 3, 4, 5, 6) to determine the most appropriate model [38–41]. The best fitting LCA model was chosen based on the most favorable fit statistics, which included Akaike's information criterion (AIC), consistent Akaike's information criterion (CAIC), and Bayesian information criterion (BIC) [33, 34]. We supplemented the fit statistics with assessments of optimal sample size per class [34], classification diagnostics, including the average posterior probability [APP] and entropy value [34–37], along with conceptual interpretability as previously done [38]. We defined appropriate model fit using APP and entropy values above 0.8 [34–37].

After identifying the optimal model, we tabulated the prevalence of different healthcare barriers and sociodemographic and health covariates by risk group. Significance testing was performed using Chi-squared (categorical variables) or adjusted Wald tests (continuous variables). We used multinomial logistic regression to predict membership in each risk group and likelihood of recurrent acute care use by risk group. We adjusted for sociodemographic, health, and insurance covariates that were significantly different among the risk groups. Our stratified analysis evaluated the association between the risk groups and recurrent acute care

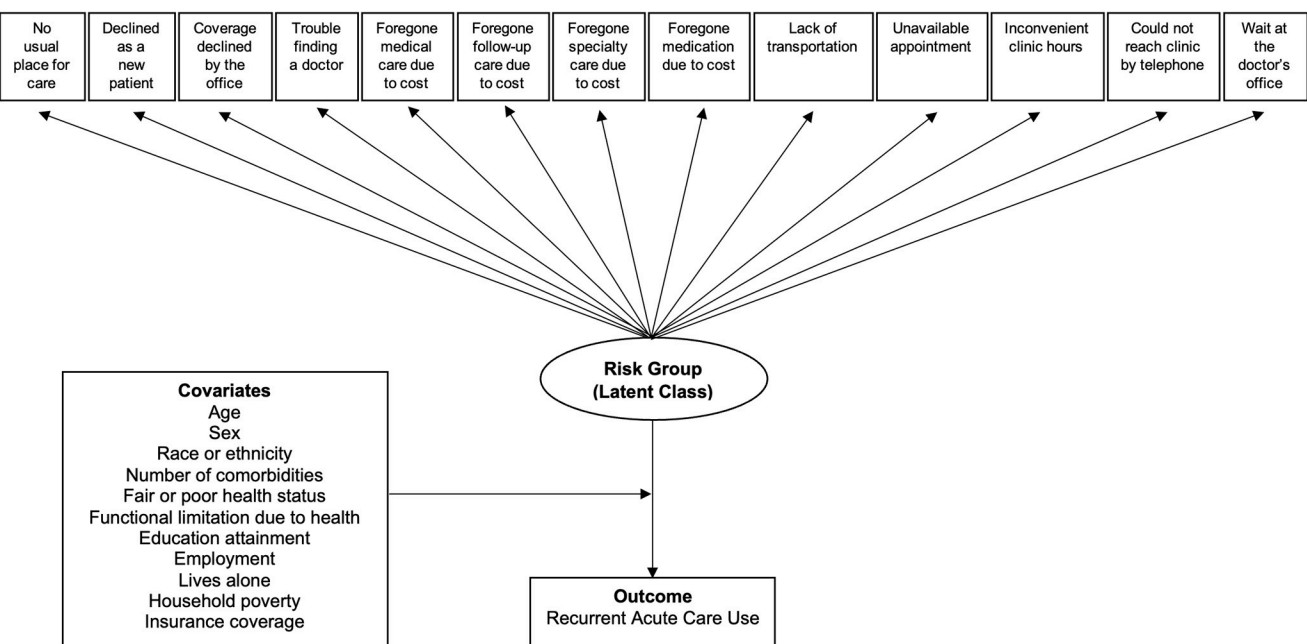

**Fig 2. Analytic model for latent class analysis.** Identification of unique risk groups (latent classes) was based on the pattern and probability of responses to the 13 questions about different healthcare barriers. The association between different risk groups and recurrent acute care use was then assessed by adjusting for covariates.

use by insurance type using multivariable logistic regression. Probabilities were obtained from the regression models.

All analyses were performed using Stata SE version 18.0 (StataCorp). Descriptive statistics and regression analyses were performed using the Stata–*svy*- command to produce nationally representative estimates. While the study population only included those with CLD, all respondents in the Sample Adult file were included in the analysis to ensure accuracy in population-level point estimates and standard errors. Fit and diagnostic statistics were obtained using unweighted data to obtain likelihood ratios. Statistical significance was defined as a 2-sided *p*-value of less than 0.05.

## Results

### Study population characteristics

The sample included 5,062 adults which provided estimates for 4,742,444 persons with CLD. Over half of the sample was female (51.5%) and non-Hispanic White (65.8%) with a median age (range) of 55 (18–85) years (Table 1). The adult population with CLD had a median number of 2 (0–9) comorbidities and suboptimal perceived health as demonstrated by rates of fair or poor health (41.4%) and the presence of a functional limitation due to health (68.2%). Most respondents had at least a high school graduate level of education (92.8%) and were not employed (57.8%). Household poverty was identified among 20.1% of respondents. About a quarter of respondents lived alone (23.2%). Half of the adult CLD population had private insurance (50.0%), followed by public insurance (23.6%), Medicare (15.3%), and no insurance (11.2%).

### Selection of the optimal latent class model

The fit statistics (AIC, CAIC, BIC) favored the models with the larger number of classes (Table 2). However, the models with five or six classes had small latent classes. The smallest APP and entropy values demonstrated acceptable model fit for models with two to four latent classes, defined as APP and entropy values greater than 0.8. As such, the best fitting latent class model identified four unique groups.

### Four risk groups identified by LCA

The risk groups were categorized as follows: group 1 represented 3,703,788 persons with CLD (78.1%) who had the lowest frequency of barriers (*minimal barriers*), group 2 represented 506,347 respondents (10.7%) with frequent experiences with healthcare unaffordability (*unaffordability*), group 3 represented 307,013 persons (6.5%) with the most organizational barriers within healthcare (*care delays*), and group 4 represented 225,296 adults (4.8%) who had frequent organizational barriers at entry to healthcare (*inability to establish care*) (Fig 3 and Table 3). In group 2 (*unaffordability*), the vast majority of respondents could not afford specialty care (78.5%). The most common barrier among individuals in group 3 (*care delays*) was inability to secure a timely appointment (90.1%). In group 4 (*inability to establish care*), the predominant healthcare barrier was being declined as a new patient (95.0%).

### Risk group characteristics and membership

Group 1 (*minimal barriers*) consisted of older adults (median age [range] 56 [18–85] years) with the lowest proportion with fair or poor health (36.9%), functional limitation due to health (64.6%), unemployment (56.4%), poverty (17.4%), and uninsurance (7.7%) (Table 1). Group 2 (*unaffordability*) included the youngest persons (median age [range] 50 [19–81]

**Table 1. Sociodemographic and health characteristics of the total sample and by risk group (N = 5,062).**

| | Total Sample | Risk Group | | | |
|---|---|---|---|---|---|
| | | Group 1 | Group 2 | Group 3 | Group 4 |
| | | Minimal Barriers | Unaffordability | Care Delays | Inability to Establish Care |
| Observations, unweighted | 5,062 | 3,953 | 540 | 328 | 240 |
| Observations, weighted | 4,742,444 | 3,703,788 | 506,347 | 307,013 | 225,296 |
| **Characteristic** | | | | | |
| Age (median, range) (year) | 55 (18–85) | 56 (18–85) | 50 (19–81) | 54 (19–85) | 52 (18–85) |
| Age groups (year) | | | | | |
| 18–34 | 13 (12–15) | 13 (12–15) | 13 (10–17) | 14 (10–19) | 16 (11–24) |
| 35–54 | 36 (34–37) | 33 (31–35) | 54 (48–60) | 37 (31–44) | 45 (37–53) |
| 55–64 | 29 (27–30) | 29 (27–31) | 24 (20–29) | 29 (24–36) | 30 (24–38) |
| 65–85 | 23 (21–24) | 25 (24–27) | 9 (6–14) | 20 (15–26) | 8 (5–13) |
| Female | 52 (50–53) | 50 (48–52) | 57 (52–63) | 55 (48–62) | 65 (56–73) |
| Race or ethnicity | | | | | |
| NH White | 66 (64–68) | 66 (64–68) | 64 (59–70) | 61 (55–67) | 73 (67–79) |
| Hispanic | 18 (16–19) | 17 (15–19) | 21 (17–26) | 23 (18–29) | 14 (10–19) |
| NH Black | 8 (8–9) | 9 (8–10) | 8 (6–11) | 9 (6–12) | 6 (4–10) |
| NH Asian | 5 (4–6) | 6 (5–7) | 3 (1–5) | 2 (1–5) | 4 (2–9) |
| NH AIAN | 1 (1–2) | 1 (1–2) | 1 (0–1) | 1 (1–3) | 1 (0–3) |
| NH Other | 2 (2–3) | 2 (2–3) | 4 (2–8) | 4 (2–8) | 2 (1–4) |
| Comorbidities (median, range)[a] | 2 (0–9) | 2 (0–9) | 3 (0–9) | 3 (0–9) | 3 (0–9) |
| Fair or poor health (n = 5,056) | 41 (40–43) | 37 (35–39) | 60 (54–66) | 62 (55–68) | 55 (46–63) |
| Functional limitation due to health (n = 5,056) | 68 (66–70) | 65 (63–67) | 80 (75–84) | 84 (78–89) | 85 (78–89) |
| Less than high school graduate level education (n = 5,036) | 7 (6–8) | 7 (6–8) | 10 (7–13) | 8 (6–13) | 9 (5–15) |
| Not employed (n = 5,052) | 58 (56–60) | 56 (54–58) | 62 (56–67) | 61 (53–67) | 71 (63–77) |
| Household poverty[b] (n = 4,791) | 20 (19–22) | 17 (16–19) | 31 (26–36) | 26 (21–32) | 35 (28–43) |
| Lives alone | 23 (22–25) | 23 (21–24) | 24 (20–29) | 28 (23–34) | 26 (20–32) |
| Insurance (n = 4,924) | | | | | |
| No insurance | 11 (10–12) | 8 (7–9) | 39 (33–45) | 7 (4–11) | 16 (11–23) |
| Public insurance | 24 (22–25) | 22 (21–24) | 19 (15–23) | 38 (31–44) | 39 (32–48) |
| Medicare | 15 (14–17) | 16 (14–17) | 15 (11–19) | 15 (10–21) | 14 (9–20) |
| Private insurance | 50 (48–52) | 55 (52–57) | 28 (23–34) | 41 (34–48) | 31 (24–39) |

Source: National Health Interview Survey, 2011–2017

Abbreviations: NH, Non-Hispanic; AIAN, American Indian or Alaskan Native

Denominators are listed for each healthcare barrier if different from the total number of observations in the sample.

All estimates are derived from a weighted sample and reported as proportions (%) with 95% confidence intervals.

Comparisons across risk groups were performed using adjusted Wald tests (continuous variable) or Chi-squared tests (categorical variables). All variables were significantly different by risk group except for education (p = 0.1806) and living alone (p = 0.1668).

[a] Comorbidities is a count variable which includes any of the following: cardiovascular disease, chronic obstructive pulmonary disease, asthma, hypertension, kidney disease, diabetes, obesity, arthritis, and history of cancer.

[b] Household poverty is based on reported annual household income below federal poverty level

years) and had the largest proportion of uninsured respondents (38.9%) compared to the other risk groups. Group 3 (*care delays*) was mostly insured (93.1%), had a higher rate of fair or poor health (61.6%), and included a larger proportion of respondents who identified as non-White, including Hispanics (23.3%), compared to other groups. Group 4 (*inability to establish care*) was predominantly female (65.1%) with more functional limitation due to

**Table 2. Model selection using fit and diagnostic statistics.**

| Model (number of classes) | Smallest Class Count (n) | Smallest Class Size (%) | AIC | CAIC | BIC | Smallest APP | Entropy |
|---|---|---|---|---|---|---|---|
| 1 | 5,062 | 100.0 | 40406.85 | 40504.73 | 40491.73 | - - | - - |
| 2 | 1,030 | 20.35 | 34801.42 | 35004.71 | 34977.71 | 0.95 | 0.86 |
| 3 | 460 | 9.09 | 33928.15 | 34236.86 | 34195.86 | 0.89 | 0.89 |
| 4 | 191 | 3.77 | 33456.74 | 33870.86 | 33815.86 | 0.86 | 0.86 |
| 5 | 146 | 2.88 | 33069.61 | 33589.15 | 33520.15 | 0.83 | 0.89 |
| 6 | 141 | 2.79 | 32963.92 | 33588.87 | 33505.87 | 0.67 | 0.81 |

Source: National Health Interview Survey, 2011–2017

Abbreviations: AIC, Akaike's information criterion; CAIC, consistent Akaike's information criterion; BIC, Bayesian information criterion; APP, average posterior probability

All estimates are derived from an unweighted sample to obtain likelihood ratios.

health (84.6%), unemployment (70.6%), poverty (35.3%), and public health insurance enrollees (39.2%).

After adjusting for sociodemographic and health characteristics, we found that membership in group 2 (*unaffordability*) was mainly insurance-driven (Table 4). Compared to the uninsured, persons in group 2 (*unaffordability*) were significantly more likely to have Medicare (adjusted rate ratio [aRRR], 0.26; 95% CI, 0.16–0.42), followed by private (aRRR, 0.14; 95% CI, 0.09–0.21) and public insurance (aRRR 0.12; 95% CI, 0.08–0.19). The most salient characteristics that contributed to membership into group 3 (*care delays*) were health-related, specifically fair or poor health (aRRR, 1.89; 95% CI, 1.29–2.77) and functional limitation due to health (aRRR, 2.54, 95% CI, 1.57–4.13). Notably, Hispanics were significantly more likely to be in group 3 (*care delays*) (aRRR, 1.62; 95% CI, 1.11–2.35). Membership in group 4 (*inability to establish care*) was also significantly associated with functional limitation due to health (aRRR, 2.59; 95% CI, 1.54–4.36) although female sex (aRRR, 1.84; 95% CI, 1.24–2.74, P = 0.003) was uniquely associated with this group.

## Association between group membership and recurrent acute care use

Group 4 (*inability to establish care*) had the highest likelihood (adjusted odds ratio [aOR], 1.85; 95% CI, 1.23–2.79) of recurrent acute care use (Table 5 and Fig 4). Our stratified analysis by insurance demonstrated similar results for recurrent acute care use and risk group (Fig 5). Group 4 (*inability to establish care*) had the highest probability of recurrent hospital and/or ED use compared to the other groups for all insurance types. Those with public insurance had the highest likelihood of recurrent acute care use compared to the other insurance types.

## Discussion

In this population-based study representative of US adults with CLD, we uncovered four unique risk groups based on self-reported healthcare barriers using LCA. This study built upon previous work [39] that highlighted socioeconomic and health vulnerabilities among US adults with CLD by identifying unique risk groups that shared different sociodemographic and health characteristics and risks of recurrent acute care use. This study provides new evidence and practice and policy implications.

First, our results highlight respondents in group 4 (*inability to establish care*) who are at particularly high risk of recurrent acute care use. Compared to the larger CLD population in group 1 (*minimal barriers*), respondents with the highest frequency of organizational barriers

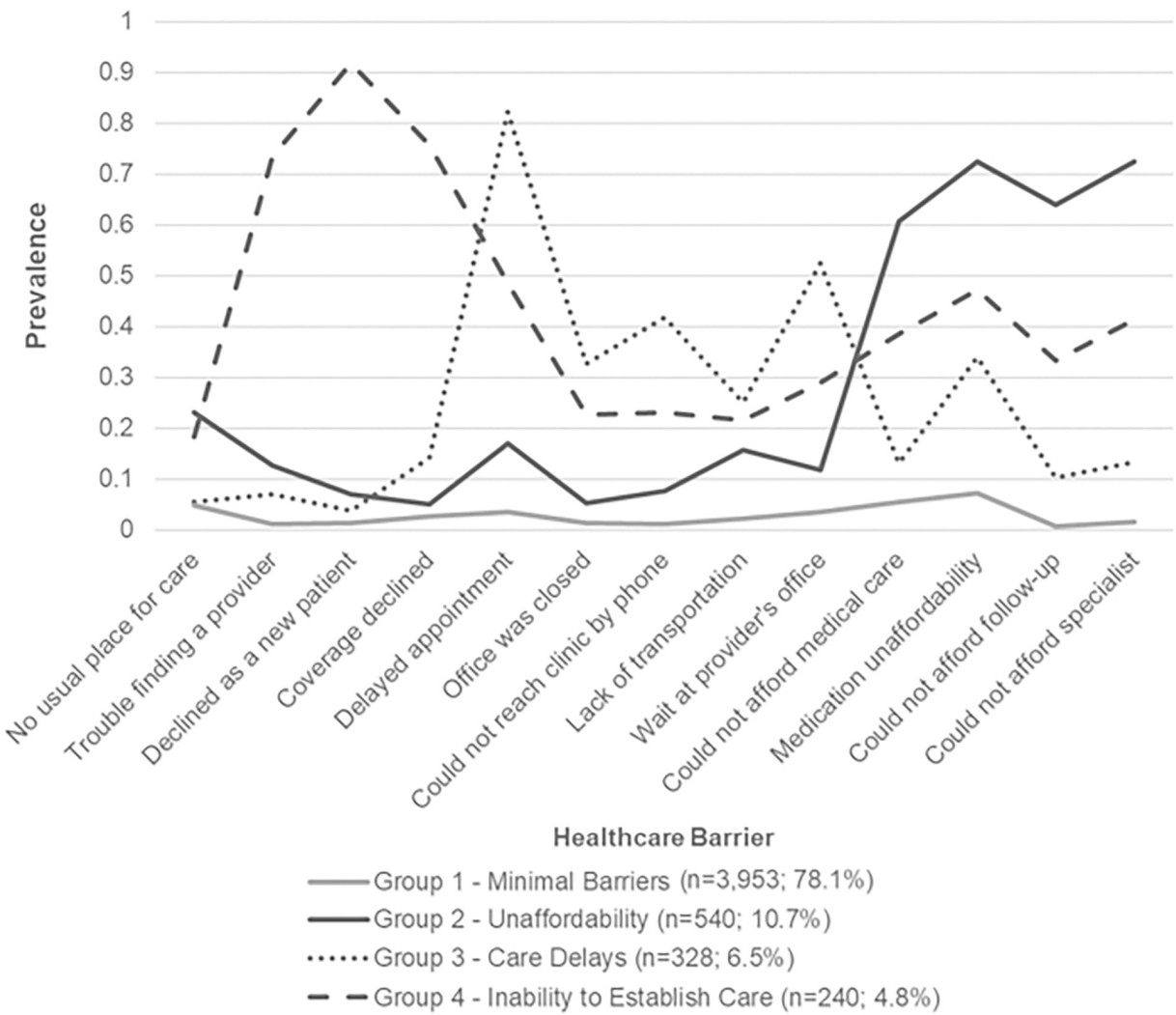

**Fig 3. Prevalence of healthcare barriers in the four group latent class model (N = 5,062).** Source: National Health Interview Survey, 2011–2017. Weighted prevalence is reported for each risk group (latent class) on the y-axis. Specific barriers are included on the x-axis. The largest group, *minimal barriers* (light solid line), included a sample of 3,953 (weighted 3,703,788) adults with chronic liver disease who had the lowest frequency of healthcare barriers. The second largest group, *unaffordability* (dark solid line), included 540 (weighted 506,347) adults who reported frequent challenges affording healthcare. The third largest group, *care delays* (dotted line), consisted of 328 (weighted 307,013) adults who reported having delays in care due to organizational barriers within the healthcare system. The fourth group, *inability to establish care* (dash line), included 240 (weighted 225,296) adults who reported having frequent barriers at entry to healthcare.

at entry to healthcare were 85% more likely to have recurrent acute care use, an estimate that was approximately 35% greater than groups 2 (*unaffordability*) and 3 (*care delays*). Females were uniquely associated with membership in group 4 (*inability to establish care*), which could be related to patient preferences for gender-concordant care [40] in the setting of a predominantly male physician workforce in the US [41]. Future efforts that aim to reduce potentially avoidable recurrent hospital-based care may have a larger effect by prioritizing respondents, particularly females, in group 4 (*inability to establish care*).

Our stratified analysis by insurance type revealed that the highest risk of recurrent acute care use occurred in the setting of public health insurance. This finding resonates with a meta-analysis that revealed a two- and three-fold lower likelihood of successful securement of

**Table 3. Prevalence of health care barriers in the total sample and by risk group (N = 5,062).**

| | Total Sample | Risk Group | | | |
| --- | --- | --- | --- | --- | --- |
| | | Group 1 | Group 2 | Group 3 | Group 4 |
| | | Minimal Barriers | Unaffordability | Care Delays | Inability to Establish Care |
| Class Prevalence (%) | 100.0 | 78.1 | 10.7 | 6.5 | 4.8 |
| Observations, unweighted | 5,062 | 3,953 | 540 | 328 | 240 |
| Observations, weighted | 4,742,444 | 3,703,788 | 506,347 | 307,013 | 225,296 |
| **Health Care Barrier** | | | | | |
| No usual place for care (n = 5,039) | 7.5 (6.6–8.6) | 5.0 (4.2–5.9) | 24.1 (19.4–29.6) | 5.0 (2.5–9.7) | 18.9 (13.2–26.4) |
| Trouble finding a provider (n = 5,035) | 6.3 (5.4–7.2) | 1.2 (0.8–1.8) | 13.8 (10.3–18.3) | 7.3 (4.2–12.6) | 77.9 (70.2–84.1) |
| Declined as a new patient (n = 5,029) | 6.5 (5.6–7.5) | 1.6 (1.1–2.3) | 7.9 (4.9–12.3) | 3.1 (1.4–6.8) | 95.0 (90.7–97.4) |
| Declined medical coverage (n = 5,032) | 7.1 (6.2–8.0) | 2.9 (2.2–3.6) | 5.3 (3.4–8.1) | 13.7 (9.4–19.4) | 78.9 (72.0–84.5) |
| Delayed appointment (n = 5,028) | 12.2 (11.1–13.3) | 3.7 (2.9–4.6) | 18.7 (14.9–23.3) | 90.1 (85.4–93.5) | 50.4 (42.3–58.5) |
| Office was closed when one could get there (n = 5,023) | 4.8 (4.2–5.6) | 1.5 (1.1–2.1) | 5.1 (3.3–7.9) | 36.6 (30.3–43.4) | 23.9 (18.2–30.7) |
| Could not reach provider's office by phone (n = 5,028) | 5.5 (4.8–6.4) | 1.2 (0.8–1.8) | 7.9 (5.3–11.5) | 48.6 (41.8–55.6) | 24.0 (18.3–30.9) |
| Lack of transportation (n = 5,028) | 6.1 (5.4–7.0) | 2.4 (2.0–3.0) | 16.2 (12.5–20.8) | 28.3 (22.3–35.2) | 21.9 (16.4–28.6) |
| Wait at the provider's office (n = 5,026) | 8.8 (7.9–9.9) | 3.7 (3.0–4.6) | 12.3 (9.2–16.3) | 58.6 (51.6–65.3) | 29.7 (22.9–37.5) |
| Could not afford recommended medical care (n = 5,061) | 13.5 (12.2–14.8) | 6.0 (5.0–7.1) | 63.4 (57.9–68.5) | 12.0 (8.2–17.3) | 36.9 (29.5–44.9) |
| Medication unaffordability (n = 5,027) | 17.8 (16.5–19.2) | 7.7 (6.7–8.9) | 75.3 (70.0–79.8) | 34.1 (27.8–41.1) | 46.8 (38.7–55.1) |
| Could not afford recommended follow-up (n = 5,027) | 9.5 (8.5–10.5) | 0.6 (0.3–1.0) | 69.4 (63.5–74.8) | 9.8 (6.5–14.6) | 33.2 (26.2–40.9) |
| Could not afford recommended specialty care (n = 5,026) | 11.7 (10.5–12.9) | 1.5 (0.9–2.4) | 78.5 (73.5–82.7) | 13.4 (9.3–19.1) | 40.6 (32.9–48.9) |

Source: National Health Interview Survey, 2011–2017

All estimates are from a weighted sample.

Total sample includes 5,062 respondents which represents an estimated 4,742,444 persons with chronic liver disease.

Denominators are listed for each healthcare barrier if different from the total number of observations in the sample.

Prevalence is reported as proportions (%) with 95% confidence intervals.

Comparisons across risk groups using Chi-squared tests were all statistically significant.

Shaded healthcare barriers represent placement into distinct risk groups according to frequency and interpretability.

medical appointments for primary and specialty care, respectively, for Medicaid compared with private insurance [42]. Future interventions that aim to mitigate organizational barriers at entry to healthcare for persons with CLD are encouraged to evaluate insurer-level policies and advocate to expand the provider network for state-sponsored insurance plans.

Second, group 3 (*care delays*) demonstrated that untimely receipt of medical care, despite having health insurance, was associated with a 50% higher likelihood of recurrent hospitalization or ED usage compared to group 1 (*minimal barriers*) similar to prior studies [19, 43]. We found a greater proportion of individuals who identified as non-White in group 3 (*care delays*), which resonates with previously reported trends in racial and ethnicity disparities in receipt of timely medical care [19]. As membership in group 3 (*care delays*) was significantly associated with more comorbidities, functional limitation due to health, and fair or poor health status, this group may be capturing individuals with more complex healthcare needs that are inadequately addressed by current system-level resources, including limited availability of appointments as shared amongst 90% of respondents in this group.

Third, this study identified approximately 11% of the US population with CLD that could be categorized in group 2 (*unaffordability*) consistent with prior studies [14, 44]. Our findings are unique because we found that insurance played the most significant role in predicting membership in group 2 (*unaffordability*) compared to other groups. In addition to capturing the largest group of uninsured persons in this group (38.9% uninsured), we found that

**Table 4. Association of Sociodemographic and health characteristics and risk group membership (n = 4,646).**

| Characteristic | Group 2 vs. 1 Unaffordability vs. Minimal Barriers RRR (95% CI) | P value | Group 3 vs. 1 Care Delays vs. Minimal Barriers RRR (95% CI) | P value | Group 4 vs. 1 Inability to Establish Care vs. Minimal Barriers RRR (95% CI) | P value |
|---|---|---|---|---|---|---|
| Age group (years) | | | | | | |
| 18–34 (reference) | 1.0 | | 1.0 | | 1.0 | |
| 35–54 | 1.12 (0.69–1.83) | 0.645 | 0.74 (0.45–1.23) | 0.243 | 0.72 (0.41–1.27) | 0.259 |
| 55–64 | 0.54 (0.32–0.90) | 0.017 | 0.60 (0.34–1.06) | 0.155 | 0.51 (0.27–0.94) | 0.032 |
| 65–85 | 0.26 (0.13–0.54) | < .001 | 0.50 (0.27–0.93) | 0.029 | 0.10 (0.04–0.20) | < .001 |
| Female | 1.20 (0.92–1.58) | 0.174 | 1.05 (0.77–1.42) | 0.762 | 1.84 (1.24–2.74) | 0.003 |
| Race or ethnicity | | | | | | |
| NH White (reference) | 1.0 | | 1.0 | | 1.0 | |
| Hispanic | 1.04 (0.72–1.50) | 0.823 | 1.62 (1.11–2.35) | 0.012 | 0.71 (0.47–1.07) | 0.102 |
| NH Black | 0.76 (0.45–1.29 | 0.307 | 0.98 (0.61–1.55) | 0.915 | 0.48 (0.26–0.90) | 0.021 |
| NH Asian | 0.62 (0.32–1.21) | 0.162 | 0.61 (0.22–1.63) | 0.321 | 0.87 (0.33–2.29) | 0.773 |
| NH AIAN | 0.20 (0.05–0.81) | 0.024 | 0.77 (0.32–1.83) | 0.548 | 0.36 (0.09–1.40) | 0.139 |
| NH Other | 1.05 (0.50–2.18) | 0.897 | 1.63 (0.66–3.99) | 0.286 | 0.40 (0.13–1.23) | 0.111 |
| Comorbidities[a] | 1.13 (1.05–1.22) | 0.002 | 1.14 (1.03–1.25) | 0.010 | 1.07 (0.96–1.19) | 0.203 |
| Fair or poor health | 1.76 (1.27–2.45) | 0.001 | 1.89 (1.29–2.77) | 0.001 | 1.01 (0.66–1.55) | 0.950 |
| Functional limitation due to health | 2.14 (1.47–3.11) | < .001 | 2.54 (1.57–4.13) | < .001 | 2.59 (1.54–4.36) | < .001 |
| Not employed | 0.99 (0.69–1.42) | 0.955 | 0.66 (0.43–1.00) | 0.052 | 1.29 (0.79–2.08) | 0.307 |
| Household poverty[b] | 1.33 (0.95–1.86) | 0.100 | 1.11 (0.76–1.62) | 0.600 | 1.40 (0.88–2.23) | 0.150 |
| Insurance | | | | | | |
| No insurance (reference) | 1.0 | | 1.0 | | 1.0 | |
| Public insurance | 0.12 (0.08–0.19) | < .001 | 1.57 (0.86–2.87) | 0.143 | 0.80 (0.47–1.33) | 0.396 |
| Medicare | 0.26 (0.16–0.42) | < .001 | 1.18 (0.60–2.34) | 0.629 | 0.78 (0.38–1.61) | 0.502 |
| Private insurance | 0.14 (0.09–0.21) | < .001 | 1.06 (0.56–2.02) | 0.852 | 0.40 (0.23–0.71) | 0.002 |

Source: National Health Interview Survey, 2011–2017

Abbreviations: RRR, relative risk ratio; CI, confidence interval; NH, Non-Hispanic; AIAN, American Indian or Alaskan Native

Risk group 2, 3, and 4 are compared to the reference risk group 1 (minimal barriers).

All estimates are derived from a weighted sample and reported as proportions (%) with 95% confidence intervals.

Estimates and significance testing were obtained using an adjusted multivariable multinomal logistic regression model.

[a] Comorbidities is a count variable which includes any of the following: cardiovascular disease, chronic obstructive pulmonary disease, asthma, hypertension, kidney disease, diabetes, obesity, arthritis, and history of cancer.

[b] Household poverty is based on reported annual household income below federal poverty level

Medicare, followed by private and public insurance were strongly associated with membership in group 2 (*unaffordability*) similar to previous reports about higher rates of medical indebtedness from private insurance plans with high-deductibles [45] compared to public insurance plans [30]. Interventions that target healthcare unaffordability among US adults with CLD are encouraged to first, reduce the prevalence of uninsurance in this disease population, and second, to decipher specific insurance plans for individuals' healthcare needs in the setting of their unique socioeconomic circumstances.

## Limitations

The study is limited in several ways. First, the study is a pooled, cross-sectional study that cannot draw causal inferences. Findings are meant to highlight significant associations and risk-

**Table 5. Odds ratios and predicted probabilities of recurrent acute care use by risk group.**

|  | Odds Ratio (95% CI) | P value | Predicted Probability (95% CI) |
|---|---|---|---|
| **Model** (vs. Group 1) |  |  |  |
| Group 2 (*Unaffordability*) |  |  |  |
| Unadjusted | 1.85 (1.45–2.37) | < .001 | 0.39 (0.34–0.45) |
| Adjusted | 1.48 (1.11–1.97) | 0.007 | 0.35 (0.30–0.40) |
| Group 3 (*Care Delays*) |  |  |  |
| Unadjusted | 2.21 (1.65–2.94) | < .001 | 0.44 (0.37–0.50) |
| Adjusted | 1.50 (1.07–2.11) | 0.019 | 0.35 (0.28–0.42) |
| Group 4 (*Inability to Establish Care*) |  |  |  |
| Unadjusted | 2.57 (1.85–3.58) | < .001 | 0.47 (0.39–0.55) |
| Adjusted | 1.85 (1.23–2.79) | 0.003 | 0.39 (0.31–0.47) |

Source: National Health Interview Survey, 2011–2017

Total observations in the unadjusted and adjusted model included 5,061 and 4,645 respondents, respectively.

Logistic regression model adjusted for age, sex, race or ethnicity, number of comorbidities, fair or poor health, functional limitation due to health, employment, household poverty, and insurance.

stratify previously undifferentiated groups of persons with CLD in the larger US adult population. Second, the study is unable to capture the etiology and severity of CLD. While we were unable to discern the causes and severity of CLD in our study, we adjusted for self-reported fair or poor health, which has been a validated measurement to personally assess one's overall

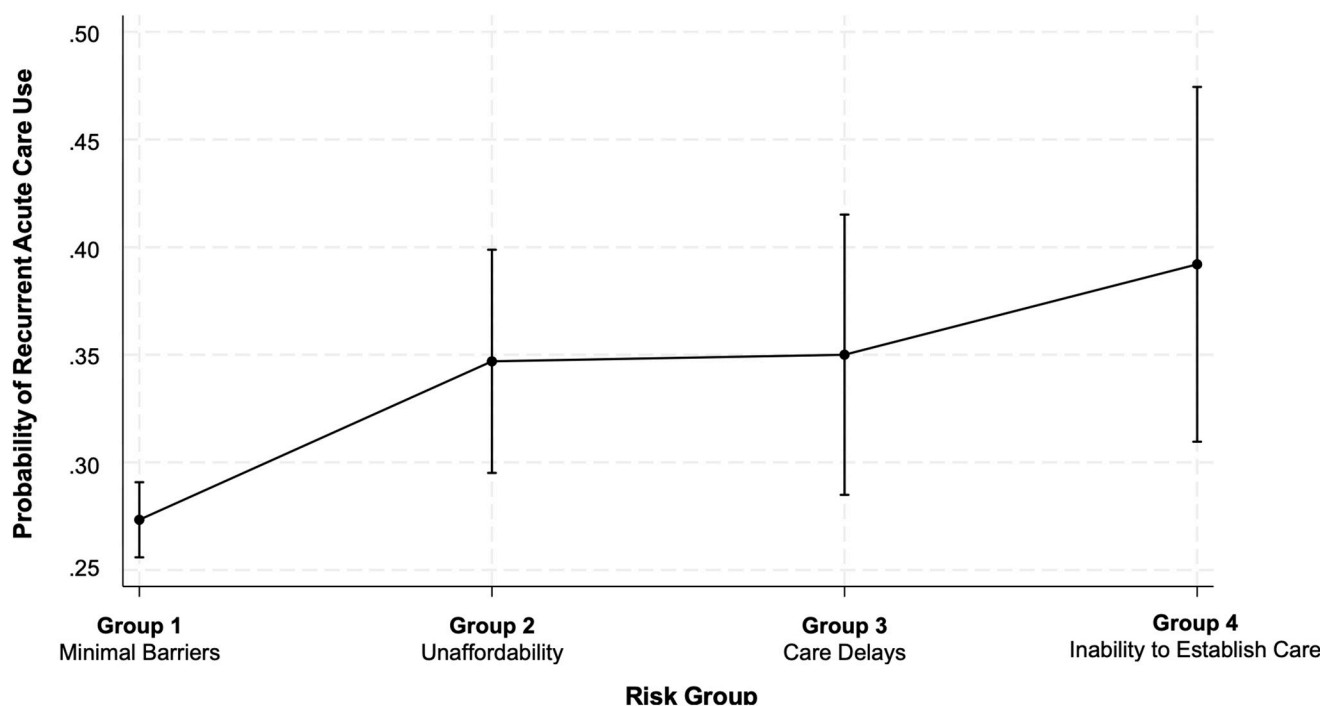

**Fig 4. Adjusted probability of recurrent acute care use by risk group.** Source: National Health Interview Survey, 2011–2017. Abbreviations: aOR, adjusted odds ratio. All estimates are from a weighted sample. Probabilities of recurrent acute care use (y-axis) are obtained from a logistic regression model (n = 4,645) that adjusted for sex, age, race or ethnicity, number of comorbidities, fair or poor health, functional limitation due to health, employment, poverty, and insurance. Adjusted probabilities of recurrent acute care use are reported for each risk group (x-axis). Compared to the reference group (*minimal barriers*), the aOR were as follows: group 2 (*affordability*) aOR, 1.48; 95% CI, 1.11–1.97; P = 0.007; group 3 (*care delays*) aOR, 1.50; 95% CI, 1.07–2.11; P = 0.019); group 4 (*inability to establish care*) aOR 1.85; 95% CI, 1.23–2.79; P = 0.003.

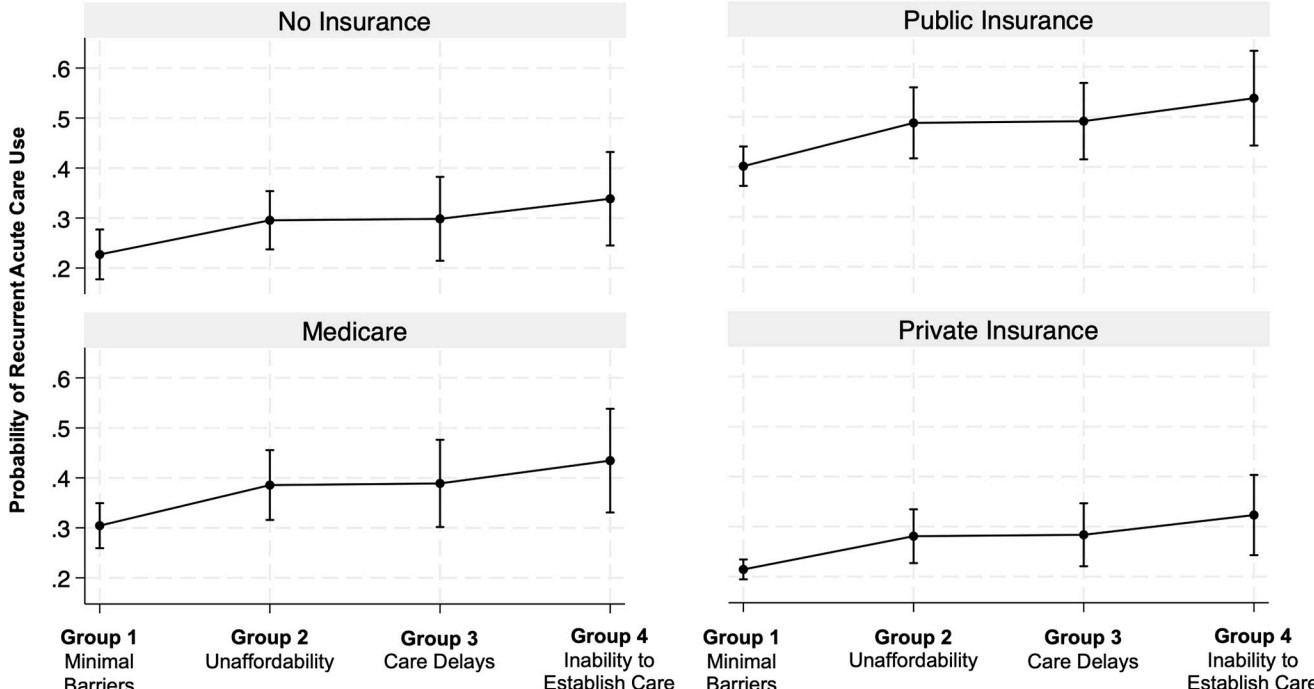

**Fig 5. Adjusted probability of recurrent acute care use by risk group and insurance.** Source: National Health Interview Survey, 2011–2017. Abbreviations: aOR, adjusted odds ratio. All estimates are from a weighted sample. Probabilities of recurrent acute care use (y-axis) are obtained from a logistic regression model (n = 4,645) that adjusted for sex, age, race or ethnicity, number of comorbidities, fair or poor health, functional limitation due to health, employment, poverty, and insurance. Adjusted probabilities of recurrent acute care use are reported for each risk group (x-axis) and by insurance type. Compared to the reference group (*minimal barriers*), the aOR were as follows: group 2 (*affordability*) aOR, 1.48; 95% CI, 1.11–1.97; P = 0.007; group 3 (*care delays*) aOR, 1.50; 95% CI, 1.07–2.11; P = 0.019); group 4 (*inability to establish care*) aOR 1.85; 95% CI, 1.23–2.79; P = 0.003.

health and severity of illnesses which may motivate one to seek medical care [24, 25]. We also adjusted for functional limitation due to health to capture the severity of health-related physical impairments as most community-dwelling persons seek acute care for concerning symptoms [22, 23]. A third limitation of this study is its lack of detailed information about health insurance plans (e.g. Medicare Advantage vs. Traditional Medicare). Inclusion of coverage details will be informative in future studies. A fourth limitation is the study period, which lacks more recent experiences about different healthcare barriers. Nonetheless, this study period provides the most currently available nationally representative data about specific healthcare barriers that were no longer captured in the NHIS starting in 2018, which remain informative in the development of future interventions to mitigate healthcare barriers for US adults with CLD. Additionally, findings from this study are based on community-dwelling, noninstitutionalized persons who knew about having CLD, which limits the generalizability of the study's findings to all US adults with CLD. Persons in institutionalized settings (e.g. nursing homes, hospitals) and correctional and treatment facilities (e.g. prisons, rehabilitation centers) or who are homeless and persons who are unaware about having CLD are not captured in this study. Therefore, interpretation this study's findings are conditional on these limitations.

## Conclusions

In conclusion, this population-based study representative of over 4.7 million US adults with CLD is the first to employ LCA to uncover four unique risk groups using self-reported,

community-based healthcare barriers. This study differentiates distinct risk groups from patterns of healthcare barriers and identifies relative risks of recurrent acute care use by these distinct groups for persons with CLD. Our results can be leveraged in future prioritization efforts that aim to reduce avoidable recurrent acute care among the most vulnerable persons with CLD, particularly those who frequently face organizational barriers at entry to healthcare.

## Supporting information

**S1 Data. Data availability statement.**
(DOCX)

## Acknowledgments

Office of Advanced Research Computing Statistical Methods and Data Analytics Group at the University of California, Los Angeles.

## Author Contributions

**Conceptualization:** Carrie R. Wong, Catherine M. Crespi, Beth Glenn, Steven-Huy B. Han, James A. Macinko, Roshan Bastani.

**Data curation:** Carrie R. Wong.

**Formal analysis:** Carrie R. Wong.

**Funding acquisition:** Carrie R. Wong.

**Investigation:** Carrie R. Wong.

**Methodology:** Carrie R. Wong, Catherine M. Crespi, Beth Glenn, James A. Macinko, Roshan Bastani.

**Supervision:** Catherine M. Crespi, Beth Glenn, James A. Macinko, Roshan Bastani.

**Validation:** James A. Macinko.

**Visualization:** Carrie R. Wong.

**Writing – original draft:** Carrie R. Wong.

**Writing – review & editing:** Carrie R. Wong, Catherine M. Crespi, Beth Glenn, Steven-Huy B. Han, James A. Macinko, Roshan Bastani.

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
