## [Decision Letter · Decision Letter 0]

9 Jul 2024

PONE-D-24-17172Distinct Risk Groups with Different Healthcare Barriers and Acute Care Use Exist in the U.S. Population with Chronic Liver DiseasePLOS ONE

Dear Dr. Wong,

Thank you for submitting your manuscript to PLOS ONE. After careful consideration, we feel that it has merit but does not fully meet PLOS ONE’s publication criteria as it currently stands. Therefore, we invite you to submit a revised version of the manuscript that addresses the points raised during the review process.

We look forward to receiving your revised manuscript.

Kind regards,

Jason T. Blackard, PhD

Academic Editor

PLOS ONE

2. In the online submission form you indicate that your data is not available for proprietary reasons and have provided a contact point for accessing this data. Please note that your current contact point is a co-author on this manuscript. According to our Data Policy, the contact point must not be an author on the manuscript and must be an institutional contact, ideally not an individual. Please revise your data statement to a non-author institutional point of contact, such as a data access or ethics committee, and send this to us via return email. Please also include contact information for the third party organization, and please include the full citation of where the data can be found.

Additional Editor Comments:

Overall, the manuscript is well-written and informative. 

Study rationale and methods are well articulated and comprehensive.

Line 224:  how is “appropriate precision” defined?  Is there a standard, accepted value / threshold?

Figure legends are needed.

Reviewers' comments:

Reviewer's Responses to Questions

**Comments to the Author**

1. Is the manuscript technically sound, and do the data support the conclusions?

Reviewer #1: Yes

Reviewer #2: Yes

Reviewer #3: Yes

2. Has the statistical analysis been performed appropriately and rigorously? 

Reviewer #1: Yes

Reviewer #2: I Don't Know

Reviewer #3: Yes

3. Have the authors made all data underlying the findings in their manuscript fully available?

Reviewer #1: Yes

Reviewer #2: Yes

Reviewer #3: Yes

4. Is the manuscript presented in an intelligible fashion and written in standard English?

Reviewer #1: Yes

Reviewer #2: Yes

Reviewer #3: Yes

5. Review Comments to the Author

Reviewer #1: This is a well written paper that describes community level risk groups and barriers to acute healthcare use among persons with CLD in the US.

The study, which analyzed self-reported data from >5000 persons responding to a National Health Interview survey, provides rich information on unique patterns of healthcare barriers at the population level among persons with CLD and how those barriers predict recurrent acute care use. The findings are important for the design of interventions, particularly those at the healthcare system level, to prevent recurrent acute care use in persons with LD. The studies strength is the large dataset of participants with robust data on barriers healthcare use and access. There are some notable limitations however, including the dates at which it was conducted (2011-2017), the lack of detailed information on chronic liver disease and its risk and the relatively high proportion of persons reporting minimal barriers to healthcare, all of which which may limit generalizability to all persons with CLD in the US.

Major comments:

Methods:

The authors should provide more details on the NHIS survey including -which states participated in the survey, how the survey was administered, whether participants were incentivized, inclusion/exclusion criteria, how representative the sample and the survey population in general was in terms of population (SES, racial groups), insurance types etc.

It is notable that a relatively high proportion of participants were NH white and almost 80% reported minimal barriers, which does not seem very generalizable.

How were the questions for this study selected? Did the authors select all question on organizational barriers, affordability, transportation etc. or just a sample. Were other questions in these categories potentially missed?

Was any data collected on ETOH use, substance use, incarceration, homelessness? Many of these are important factors at the individual level in persons with CLD and associated with health care access issues as well as frequent hospital visits. If no data was collected on these it should be mentioned in the limitations.

The sentence ‘ED visits were included in the outcome variable as ED visits within 30 days of a hospital discharge could account for up to 40% of recurrent acute care encounters’ does not make sense and needs clarifying. Is it supposed to be ‘ED visits ‘that’ were….’

For the definition of recurrent care – could this be for any cause ie. That was unrelated to liver disease. If so, this needs to be made clear.

Line 129 states ‘We selected self-reported healthcare barriers that spanned the care seeking process including organizational barriers, healthcare unaffordability, and transportation insecurity’; in reality, there was only one question about transportation so the authors may want to remove ‘transportation insecurity’ in this line.

Statistics: Most participants fell into minimal barriers group. Only a very small percent 4.8% were in the inability to establish care? Was the study adequately powered to determine the association between suboptimal access and the primary outcome of likelihood of recurrent acute care use.

Discussion:

The discussion is well written and appropriately addresses the main findings of the study. However noticeably absent is any discussion about gender and its association with care delays. Could the authors comment on why this might be?

Tables/Figures:

There are no figure legends.

Reviewer #2: In this manuscript by Wong et al, the authors use the NHIS to identify distinct sets of health care barriers and their risk of being associated with recurrent acute care use among people with self-reported CLD. This topic is important and timely, highlighting that many patients with CLD experience socioeconomic barriers. The findings are important as they identify specific barriers, which are potential targets for future interventions aimed at reducing health care disparities. Most of the limitations are related to the use of a large population based data set, which inherently lacks granularity.

Major comments:

Was CLD only self-reported? Is there any access to ICD codes that could be used to cross-check whether participants truly had liver disease?

Do you have any info on comorbidities? Like the Charlston comorbidity index? This may be a significant confounder to recurrent acute care use outcomes, regardless of SES barriers that exist.

Minor comments:

Very minor comment but consider removing decimals in table 1 and reporting whole numbers, currently looks very cluttered

I think another major finding from this is that 1/5th of all people with CLD live in poverty – I would highlight what a disadvantaged population people with CLD represent. Also that 41% are in fair or poor health or that 68% are functionally limited due to their health – these are striking numbers.

Reviewer #3: Congratulations on a well thought-out and well written article. Evaluating non-hospital based systemic factors affecting the healthcare outcomes is an area of unmet need. Identification of distinct risk groups with chronic liver disease using statistical models is novel. I have a few comments/questions/suggestions

- Affordable Care Act was passed in 2010. It is very well documented that first couple of years, the healthcare exchanges went through lots of technical challenges and enrollment was low. Would excluding first 2-3 years after passing of the Act and including more years at the tail end made your data more robust and representative of current healthcare landscape

- The screening question regarding presence of liver condition/chronic liver disease appears very broad and non-specific

- Although authors tried to improve the size of latent classes by reducing the number of groups from 6 to 4, it appears that group 2-4 still have small sample size

- For group 4, does your data allow any further analyses to determine the factors contributing to inability to establish care

6. PLOS authors have the option to publish the peer review history of their article (what does this mean?). If published, this will include your full peer review and any attached files.

Reviewer #1: No

Reviewer #2: No

Reviewer #3: No

---

## [Author Response · Author response to Decision Letter 0]

15 Aug 2024

Editor

Comment #1: 

Overall, the manuscript is well-written and informative. Study rationale and methods are well articulated and comprehensive. Line 224: how is “appropriate precision” defined? Is there a standard, accepted value/ threshold?

Response to Comment #1: 

We appreciate your comments and recognize the need to be clearer in the Methods – Statistical Analysis section. We have modified our sentence to the following: “The smallest APP and entropy values demonstrated acceptable model fit for models with two to four latent classes, defined as APP and entropy values greater than 0.8.” 

Comment #2: 

Figure legends are needed

Response to Comment #2: 

Thank you for bringing this to our attention. We have added legends to Figures 1-5.

Reviewer #1

Comment #1:

This is a well written paper that describes community level risk groups and barriers to acute healthcare use among persons with CLD in the US.

The study, which analyzed self-reported data from >5000 persons responding to a National Health Interview survey, provides rich information on unique patterns of healthcare barriers at the population level among persons with CLD and how those barriers predict recurrent acute care use. The findings are important for the design of interventions, particularly those at the healthcare system level, to prevent recurrent acute care use in persons with LD. The studies strength is the large dataset of participants with robust data on barriers healthcare use and access. There are some notable limitations however, including the dates at which it was conducted (2011-2017), the lack of detailed information on chronic liver disease and its risk and the relatively high proportion of persons reporting minimal barriers to healthcare, all of which may limit generalizability to all persons with CLD in the US.

Response to Comment #1: 

We appreciate your thoughtful comments and agree with your stated limitations. We recognize that the study period, 2011-2017, and lack of detailed information about CLD, are limitations of this study which have been included in the revised Limitations section. We selected the study period to capture different self-reported healthcare barriers after the enactment of the Affordable Care Act in 2010 and before specific healthcare barriers were no longer surveyed starting in 2018. We understand that the lack of clinical data about CLD is a limitation of this dataset. However, we sought to use NHIS data because it uniquely provides nationally representative data among adults with CLD and captures information about community-based healthcare barriers which are unavailable through health system or administrative data. While most US adults with CLD were in the minimal barriers group, this study provides helpful data about the most vulnerable groups (groups 2-4), representative of approximately 1 million US adults with CLD, which may be targeted for future interventions to alleviate specific healthcare barriers.

Comment #2:

Methods: The authors should provide more details on the NHIS survey including which states participated in the survey, how the survey was administered, whether participants were incentivized, inclusion/exclusion criteria, how representative the sample and the survey population in general was in terms of population (SES, racial groups), insurance types, etc. It is notable that a relatively high proportion of participations were NH White and almost 80% reported minimal barriers, which does not seem very generalizable.

Response to Comment #2: 

Thank you for your helpful critiques. The NHIS is conducted by the National Center for Health Statistics (NCHS) using interviewers from the Census Bureau. The NHIS is widely accepted as a nationally representative and reliable source of information for research purposes. The NHIS target population is the civilian, noninstitutionalized population residing in the US, including all 50 states and the District of Columbia, at the time of the interview. Persons in institutionalized settings, including long-term care institutions (e.g. nursing homes, hospitals), correctional facilities (e.g. prisons, jails), and US nationals residing in foreign countries are excluded from the survey. Active-duty Armed Forces personnel are also excluded from the NHIS unless at least one other family member is a noninstitutionalized civilian who is eligible for the survey. We have clarified in the Methods – Data Source section that participants are noninstitutionalized, community-dwelling persons residing in the 50 US states and the District of Columbia at the time of the interview. The survey is conducted in a face-to-face format using a standardized questionnaire within households. The NHIS does not provide any incentives or compensation for participation.

The administration of the survey is based on its study design. For the years included in this study, the NHIS survey consisted of a sample of over 300 clusters of addresses that are in well-defined geographic areas. These geographic areas consisted of a county, a small group of contiguous counties, or a metropolitan area, which are all within the boundaries of a single state. The geographic areas were classified into one of two groups within each state. Within each group classification and within each geographic area, clusters of addresses were defined using 2010 Census housing unit count information. A systematic sample of address clusters were selected independently from each group classification. The locations of the selected address clusters are where NHIS field interviewing resources are allocated for implementing the current sample design. 

The sample design is a probability design that allows representative sampling of households and noninstitutionalized groups, and the analysis of the NHIS data includes individual weights provided by the NCHS. The NHIS documentation describes the weighting process as follows: “The NHIS sampling procedure allows the creation of household- and person-level base weights. Each base weight is the inverse of the probability of selection. Roughly speaking, the base weight is the number of population units a sampled unit represents. Under ideal sampling conditions, and if 100% response occurred, a base-weighted sample total will be an unbiased estimator for the true total in the target population. In practice, however, the base weights are adjusted for nonresponse and ratio-adjusted to create final sampling weights. The final person-level weights are also adjusted accordingly to a quarterly poststratification by age/sex/race/ethnicity classes based on population estimates produced by the US Census Bureau.” More information is available here: https://ftp.cdc.gov/pub/Health_Statistics/NCHS/Dataset_Documentation/NHIS/2018/srvydesc.pdf.

The provided analytic weights permit nationally representative estimates of the civilian noninstitutionalized population that are generalizable to the eligible US population. The current study includes only adults with self-reported CLD; therefore, estimates from this sample are only applicable to the eligible US population with known CLD. Nonetheless, we recognize the limited generalizability of our study’s findings to persons in institutionalized settings (e.g. hospitals, rehabilitation settings, prisons) who are also at risk for recurrent acute care use. We have added this issue to our revised Limitations section. While the majority of participants reported having minimal barriers, a notable number of US adults with CLD, representative of approximately 1 million persons, faced frequent and unique healthcare barriers that could be potentially targeted to help reduce preventable recurrent acute care use among the most vulnerable persons with CLD.

Comment #3:

How were the questions for this study selected? Did the authors select all question on organizational barriers, affordability, transportation etc. or just a sample. Were other questions in these categories potentially missed?

Response to Comment #3: 

Before selecting questions for this study, we set 2011 as the start of the study period because we aimed to assess healthcare experiences after the enactment of the Affordable Care Act. We reviewed all questions pertaining to medical care/ access starting in 2011on the Integrated Public Use Microdata Series website where we extracted our dataset [1]. To ensure a homogenous dataset as we pooled annual cross-sectional NHIS data, we confirmed that all variables/ questions used in the study were comparable over time for all adult respondents during the study period, all questions were asked annually during the study period, and all sample adults remained in the “universe” of each question (i.e. questions for only a subgroup of sample adults were not included). We ended the study period in 2017 because specific questions about organizational barriers (e.g. “needed but couldn’t afford follow-up care”, “had trouble finding a general doctor”, “told not accepted as a new patient”, and “told health care coverage not accepted”) were no longer surveyed starting in 2018. We also selected questions that were more generalizable and directly related to potential risk of recurrent acute care use (e.g. included “needed but couldn’t afford medical care” and excluded “needed but couldn’t afford eyeglasses”) to create a parsimonious set of high-quality indicators/ healthcare barriers to perform latent class analysis. 

Comment #4:

Was any data collected on ETOH use, substance use, incarceration, homelessness? Many of these are important factors at the individual level in persons with CLD and associated with health care access issues as well as frequent hospital visits. If no data was collected on these it should be mentioned in the limitations.

Response to Comment #4:

We agree that additional individual-level factors, including substance use, incarceration, and homelessness, are associated with healthcare access and recurrent acute care use. The dataset lacks information about incarceration, homelessness, and substance abuse in the adult sample. A notable limitation is that the surveyed sample only includes noninstitutionalized persons such that persons in institutionalized settings (e.g. hospital) and correctional and treatment facilities (e.g. rehabilitation centers, prisons, halfway houses) are not captured in this study, which we have added to the revised Limitations section. Alcohol use was measured as lifetime abstainer, former, or current user. We omitted alcohol in our analysis because we found such measurements of alcohol use less clinically meaningful. 

Comment #5:

The sentence ‘ED visits were included in the outcome variable as ED visits within 30 days of a hospital discharge could account for up to 40% of recurrent acute care encounters’ does not make sense and needs clarifying. Is it supposed to be ‘ED visits ‘that’ were….’

Response to Comment #5:

We recognize the need to be clearer in this statement. We have modified the sentence to the following: “We did not limit hospital-based acute care to readmissions because up to 40% of recurrent acute care encounters within 30 days of a hospital discharge could occur in the ED without readmission.”

Comment #6:

For the definition of recurrent care – could this be for any cause ie. That was unrelated to liver disease. If so, this needs to be made clear.

Response to Comment #6:

Thank you for bringing this to our attention. We have modified that recurrent acute care use is defined as having at least two emergency department visits and/or overnight hospital admissions for any cause in the past year in the Methods – Measures section.

Comment #7:

Line 129 states ‘We selected self-reported healthcare barriers that spanned the care seeking process including organizational barriers, healthcare unaffordability, and transportation insecurity’; in reality, there was only one question about transportation so the authors may want to remove ‘transportation insecurity’ in this line.

Response to Comment #7:

Thank you for your input. We have excluded transportation insecurity from the sentence.

Comment #8:

Statistics: Most participants fell into minimal barriers group. Only a very small percent 4.8% were in the inability to establish care? Was the study adequately powered to determine the association between suboptimal access and the primary outcome likelihood of recurrent acute care use.

Response to Comment #8:

The reviewer is correct that a small percent of the sample was in the inability to establish care group. Smaller samples in our model are adequate contingent that the classes are “well-separated” [2,3]. The classification diagnostics for the 4-class model revealed adequate model fit with the smallest average posterior probability (APP) and entropy values of 0.86 (both above 0.8 which was the defined threshold). While the inability to establish care group consisted of only 4.8% of the total sample of US adults with CLD, because the full sample itself was large (N=5,062 respondents with CLD), there were 240 respondents in this subgroup, which is adequate to support regression analyses comparing it with the minimal barriers group (n=3,953 respondents). In fact, significant associations were detected for this subgroup. This subgroup represented 225,296 US adults with CLD who experienced frequent organizational barriers at entry to healthcare. 

Comment #9:

Discussion: The discussion is well written and appropriately addresses the main findings of the study. However noticeably absent is any discussion about gender and its association with care delays. Could the authors comment on why this might be?

Response to Comment #9:

Thank you for your comments. We did not discuss female (vs. male) sex and its association with the care delays group because it was not a significantly different variable after adjusting for other covariates (aRRR 1.05; 95% CI 0.77-1.42; P=0.762). We noted that female sex was a salient variable in the inability to establish care group (aRRR 1.84, 95% CI 1.24-2.74, P=0.003) and have included additional discussion about this finding in the revised manuscript. The significant association between females and the inability to establish care group may be related to patient preferences for gender-concordant care in the setting of a predominantly male physician workforce in the US.

Comment #10:

Tables/Figures: There are no figure legends.

Response to Comment #10:

We have added legends to Figures 1-5. Thank you for bringing this to our attention.

Reviewer #2

Comment #1:

In this manuscript by Wong et al, the authors use the NHIS to identify distinct sets of health care barriers and their risk of being associated with recurrent acute care use among people with self-reported CLD. This topic is important and timely, highlighting that many patients with CLD experience socioeconomic barriers. The findings are important as they identify specific barriers, which are potential targets for future interventions aimed at reducing health care disparities. Most of the limitations are related to the use of a large population based data set, which inherently lacks granularity.

Was CLD only self-reported? Is there any access to ICD codes that could be used to cross-check whether participants truly had liver disease?

Response to Comment #1:

Thank you for your comments. We agree that the use of a population-based dataset like the NHIS is limited as it uses self-reported information and is unable to cross-check with clinical data to confirm diagnoses of CLD. Nonetheless, the questions about CLD were conditional on having a doctor or other health professional informing the participant about having CLD. In contrast to using administrative data or electronic health records (EHR) which include more clinical information, we sought to use NHIS data because it uniquely provides nationally representative data among US adults with CLD in the community setting. The NHIS offers the opportunity to learn about healthcare experiences in the outpatient setting, which are not routinely captured in administrative or EHR data.

Comment #2:

Do you have any info on comorbidities? Like the Charlston comorbidity index? This may be a signifi

---

## [Decision Letter · Decision Letter 1]

12 Sep 2024

Distinct Risk Groups with Different Healthcare Barriers and Acute Care Use Exist in the U.S. Population with Chronic Liver Disease

PONE-D-24-17172R1

Dear Dr. Wong,

We’re pleased to inform you that your manuscript has been judged scientifically suitable for publication and will be formally accepted for publication once it meets all outstanding technical requirements.

Kind regards,

Jason T. Blackard, PhD

Academic Editor

PLOS ONE

Additional Editor Comments (optional):

None

Reviewers' comments:

None

Reviewer's Responses to Questions

**Comments to the Author**

1. If the authors have adequately addressed your comments raised in a previous round of review and you feel that this manuscript is now acceptable for publication, you may indicate that here to bypass the “Comments to the Author” section, enter your conflict of interest statement in the “Confidential to Editor” section, and submit your "Accept" recommendation.

Reviewer #3: All comments have been addressed

2. Is the manuscript technically sound, and do the data support the conclusions?

Reviewer #3: Yes

3. Has the statistical analysis been performed appropriately and rigorously? 

Reviewer #3: Yes

4. Have the authors made all data underlying the findings in their manuscript fully available?

Reviewer #3: Yes

5. Is the manuscript presented in an intelligible fashion and written in standard English?

Reviewer #3: Yes

6. Review Comments to the Author

Reviewer #3: (No Response)

7. PLOS authors have the option to publish the peer review history of their article (what does this mean?). If published, this will include your full peer review and any attached files.

Reviewer #3: No

---

## [Editor Report · Acceptance letter]

21 Oct 2024

PONE-D-24-17172R1 

PLOS ONE

Dear Dr. Wong, 

I'm pleased to inform you that your manuscript has been deemed suitable for publication in PLOS ONE. Congratulations! Your manuscript is now being handed over to our production team.

Kind regards, 

on behalf of

Dr. Jason T. Blackard 

Academic Editor

PLOS ONE